# THE POWER OF DATA: HOW LSTMS OUTSHINE DISEASE PROGRESSION MODELING WITH TWO SIMPLE MECHANISMS

## ABSTRACT

Much of prior efforts have focused on Disease Progression Modeling (DPM) using Electronic Health Records (EHRs). EHRs, however, present significant challenges for deep learning models such as Long Short-Term Memory (LSTM), Variational Recurrent Neural Networks (VRNN), and Transformer due to the inherent complexities and *variabilities* within the data. Effectively addressing these variabilities is crucial for improving the performance and interpretability of such models. In this work, we propose *two mechanisms* to tackle key variabilities in EHR data: a **"bi-directional"** mechanism to account for the need to infer the underlying physical state in both forward and backward directions, and a **"time-aware"** mechanism to address *irregular time intervals* between consecutive events. We *theoretically validate and empirically evaluate* the impact of these two mechanisms across three state-of-the-art deep learning models in three distinct healthcare systems. Our results showed that the influence of the two mechanisms—bidirectionality and time-awareness—surpasses the differences between specific deep learning models. Across all three models, the performance hierarchy consistently follows: *Bidirectional & Time-Aware > Time-Aware > Bidirectional > Original model*, across all three healthcare systems. Notably, the Bidirectional Time-Aware LSTM matches or exceeds the performance of the corresponding VRNN and Transformer models in every system tested.

## 1 INTRODUCTION

Electronic Health Records (EHRs) represent a comprehensive and large-scale repository of temporal health data for patients. Their widespread adoption in healthcare systems has driven the development of deep learning models designed to model patient histories and predict health risks (Marlin et al., 2012; Choi et al., 2016a; Zhou et al., 2013; Choi et al., 2016b). In this work, we focus on the task of *Disease Progression Modeling (DPM)* , which monitors the disease developing process and predicts future risks based on patients' historical information, and we focus on a specific disease *septic shock*, which is life-threatening organ dysfunction and has an extremely high mortality rate. DPM plays a crucial role in predicting the trajectory of a patient's condition over time, enabling clinicians to make more informed decisions about treatment and intervention.(Cook & Bies, 2016), By analyzing longitudinal EHR data, DPM aim to capture the complex, nonlinear patterns in patient vitals, lab results, and other clinical features, thereby forecasting the future state of a disease. This is particularly important in critical care settings, where early detection of severe conditions like septic shock can be life-saving (Singer et al., 2016).

A large amount of recent works have applied various deep learning models for DPM (Kim & Chi, 2018; Zhang et al., 2019a; Zhang, 2019; Zhang et al., 2017b). Recurrent Neural Networks (RNNs) are among the most widely researched deep learning models for processing sequential data like EHRs (Choi et al., 2016a; Esteban et al., 2016; Lipton et al., 2015; Zhou et al., 2013). Extensions of RNNs, such as **Long Short-Term Memory (LSTM)**, are specifically designed to capture long-term dependencies within patients' records over extended periods (Sundermeyer et al., 2012; Wells et al., 2013; Men et al., 2021; Maragatham & Devi, 2019). Similarly, **Variational RNN (VRNNs)** (Chung et al., 2015; Khoshnevisan & Chi, 2020; Jun et al., 2020) have shown to be effective at addressing missingness and capturing complex conditional and temporal dependencies in EHRs(Zhang et al.,

2017a; Mulyadi et al., 2020). More recently, **Transformers** (Vaswani et al., 2017) leverage a self-attention mechanism to capture long-range dependencies between medical events, making them highly effective for modeling temporal patterns in EHRs (Li et al., 2022; 2020).

EHRs, however, pose numerous challenges for deep learning models due to the *inherently complex variabilities*. Deep reasoning beyond these variabilities is the key to understand, study and improve the outcomes of a disease, and hence serves a better medical care delivery to public health. While the standard LSTM, VRNN, and Transformer models have demonstrated considerable success, notable challenges persist when applying them to model EHRs: one is that these models exclusively process input sequences in a *one-directional forward* manner; the other is that they do not account for *irregular time intervals* between consecutive events. In this work, we investigated *two mechanisms* to tackle the two key variabilities in EHR data: a **"bi-directional"** mechanism to account for the need to infer the underlying physical state in both forward and backward directions and a **"time-aware"** mechanism to address *irregular time intervals* between consecutive events.

**Bidirectional Nature:** *The bidirectional nature* of EHRs is critical to not only capture the context preceding a specific time step, as seen in standard LSTM and VRNN, but also the future context that follows. For example, one major challenge associated with early prediction of septic shock is the subtle but fast progression at early stage: only minor changes are reflected on white blood cells and body temperature at early stage (Kumar et al., 2006). Besides, the indicators of sepsis are non-specific, such as infection or fast heart rate, and patients with such symptoms are highly likely to progress to other diseases. Thus, considering both historical and forthcoming information allows for a more comprehensive understanding of the patient's condition at a given timestamp. Bidirectional models such as bi-LSTM (Huang et al., 2015) or advanced Transformer model such as BERT(Devlin et al., 2018) process sequences in both forward and backward directions, enhancing the contextual understanding of the data.

**Time-Aware for Irregular Time Intervals:** Measurements in EHRs are often acquired with *irregular intervals*. For example, when a patient is under sever conditions,, events tend to be recorded more frequently than during periods of relative stability. These irregular time intervals can reveal important insights into a patient's health status and potential impending conditions. Therefore, it is essential to take into account the time intervals between temporal events to capture latent progressive patterns of a disease. There have been several previous works on handling the time irregularity (Baytas et al., 2017; Pham et al., 2016; Choi et al., 2016a; Che et al., 2017), e.g, Time-aware LSTM (T-LSTM) (Baytas et al., 2017) transforms time intervals into weights to adjust the memory passed from previous moments.

Previous studies have only investigated one of the two mechanisms—either Bidirectionality or Time-awareness individually— combined with one of three deep learning models, LSTM, VRNN, or Transformer. No research has comprehensively examined the combined impact of both mechanisms across all three models. In this work, we address this critical gap by *theoretically validating and empirically evaluating* the integration of both Bidirectionality and Time-awareness across three state-of-the-art deep learning models in three distinct healthcare systems. Specifically, we investigate the effectiveness of two key mechanisms—Bidirectionality and Time-awareness—and their combination, Bidirectional Time-Aware—in improving the performance of foundational deep learning models for the crucial task of early septic shock prediction. **Sepsis** constitutes a critical condition characterized by life-threatening organ dysfunction (Singer et al., 2016) and stands as a prominent cause of mortality in the United States. The most severe outcome of sepsis, known as *septic shock*, is associated with a mortality rate that can reach up to 50% (Martin et al., 2003), along with a growing annualized incidence (Dellinger et al., 2008). Timely diagnosis and intervention could potentially prevent up to 80% of sepsis-related deaths (Kumar et al., 2006). Early prediction of septic shock is challenging due to the presence of ambiguous symptoms and subtle physiological responses (Kumar et al., 2006). Additionally, similar to cancer, sepsis encompasses diverse disease etiologies spanning a broad spectrum of syndromes. Various patient groups may exhibit markedly distinct symptoms, adding complexity to the understanding and diagnosis of sepsis (Tintinalli et al., 2011). Due to the nuanced nature of these subtle progressions, variables in the pre-shock stage may either be infrequently measured or remain unmeasured altogether. Consequently, it is paramount to incorporate both bi-directional information and irregular time intervals into consideration for a comprehensive understanding and early prediction of septic shock.

We leverage EHRs collected from three large medical systems: *Christiana Care Health System (CCHS)* in Newark, Delaware and ICU visits of patients admitted to Beth Israel Deaconess Medical Center in Boston, Massachusetts (2001-2012), *MIMIC-III* (Johnson et al., 2016), and patient data from Mayo Clinic, *Mayo*. Our experimental findings across these three real-world EHR datasets demonstrate that the Bidirectional Time-Aware mechanism consistently enhances performance for LSTM, VRNN, and Transformer models. Incorporating both Bidirectionality and Time-awareness leads to more accurate early predictions of septic shock, particularly in models like the LSTM, which can effectively capture the complexity of subtle symptom progression and irregular time intervals. Across all three datasets, the Bi-T model exhibits the highest performance. Our contributions are:

- This work offers a simple yet effective approach by integrating bidirectional and time-aware mechanisms across three neural network architectures. To our knowledge, the proposed Bi-T-LSTM, Bi-T-VRNN, and Bi-T-Transformer represent one of the first attempts to combine these two mechanisms in deep learning models.

- We provide theoretical insights for each of the three deep learning models, explaining why the proposed mechanisms would improve performance compared to the original models. Our results showed that the influence of the two mechanisms—bidirectionality and time-awareness—surpasses the differences between specific deep learning models.

- Our results include a comparative evaluation of the standard LSTM, state-of-the-art VRNN, and Transformer models, assessing the impact of different configurations, including those with and without the proposed mechanisms across three large healthcare datasets: MIMIC-III, CCHS, and Mayo.

The remainder of the paper is organized as follows: In Section II, we elucidate the integration of our proposed bidirectional and time-aware mechanisms with standard LSTM, VRNN, and Transformer models. Section III provides details on our three datasets, the prediction task, hyperparameter tuning, and the evaluation metrics employed. Section IV presents our results, while Section V delves into related work. Finally, our conclusions are presented in the concluding section.

## 2 The Mechanism of Time embedding: $\Delta T$

Our dataset consists of multi-variate irregular time series data and can be represented as $\boldsymbol{X} = \{\boldsymbol{x}_1, \boldsymbol{x}_2, \ldots, \boldsymbol{x}_N\}$, where $N$ denotes the total number of hospital visits. Each hospital visit $\boldsymbol{x}_k$ consists of a sequence of clinical events over time: $\boldsymbol{x}_k = \{\boldsymbol{x}_k^1, \ldots, \boldsymbol{x}_k^{T_k}\}$, where $\boldsymbol{x}_k^t$ represents the patient's clinical measurements at time step $t$ during visit $k$. Specifically, $\boldsymbol{x}_k^t \in \mathbb{R}^D$, where $D$ is the number of recorded features at each event, and $T_k$ is the number of events during visit $k$, which varies across visits. Each sequence $\boldsymbol{x}_k$ is associated with event-level labels $\boldsymbol{y}_k = \{y_k^1, \ldots, y_k^{T_k}\}$, where $y_k^t = 1$ indicates that the patient is in septic shock at time step $t$, and $y_k^t = 0$ otherwise. The objective of this work is to predict the label $y_k^{t+1}$ for the next event given the sequence of clinical events up to time $t$, i.e., $\boldsymbol{x}_k^1, \boldsymbol{x}_k^2, \ldots, \boldsymbol{x}_k^t$ for each visit $k$. For simplicity, we omit the index $k$ where it does not lead to ambiguity.

### 2.1 Incorporating Time Intervals ($\Delta t$) in LSTM

We describe the incorporation of time intervals, $\Delta t$, in the general case first and then illustrate the mathematical proof using LSTM as an example. A similar proof for combining $\Delta t$ with VRNN and transformers is available in the appendix.

In a standard LSTM setup, the model implicitly assumes a uniform unit time of $\Delta t = 1$ between consecutive events. This implies that transitions in the hidden states and updates to the cell states are designed under the premise that each time step represents an equal interval of temporal progression, represented as $h^t = f(W_h^h h^{t-1} + W_h^x x^t + b_h)$. However, in real-world applications such as EHR data, the time intervals $\Delta t$ between consecutive observations vary significantly, ranging from minutes to hours depending on the patient's condition.

This variability introduces issues in standard LSTMs. If $\Delta t$ is less than 1 (e.g., closely spaced events), the LSTM might overestimate the significance of small changes. If $\Delta t$ is greater than 1, the LSTM may underestimate significant changes.

To address this, time embeddings are introduced, making the LSTM aware of actual time intervals. Let $\Delta \tilde{t}$ represent the time difference between consecutive observations, $t - 1$ and $t$. This interval is embedded using a time embedding matrix $E_t$, with $e^t = E_t(\Delta \tilde{t})$. The embedding $e^t$ is concatenated with the input features $x^t$, resulting in an augmented input $\tilde{x}^t = \text{concat}(x^t, e^t)$. The LSTM then becomes $h^t = f(W_h^h h^{t-1} + W_x^h \tilde{x}^t + b_h)$.

### 2.1.1 LSTM GATE UPDATES WITH TIME-AWARENESS

The hidden state transition is refined by incorporating all LSTM gates (input, forget, output, and cell state). These gates take into account the varying time intervals by using a decay function $\gamma(\Delta \tilde{t}) = \exp(-\alpha \Delta \tilde{t})$, where $\alpha$ is a learnable decay rate.

**Forget Gate:** Controls how much of the previous cell state $c^{t-1}$ is retained:

$$f^t = \sigma \left( W_x^f \tilde{x}^t + U_h^f h^{t-1} \cdot \gamma(\Delta \tilde{t}) + b_f \right)$$

**Input Gate:** Determines how much new information is added:

$$i^t = \sigma \left( W_x^i \tilde{x}^t + U_h^i h^{t-1} \cdot \gamma(\Delta \tilde{t}) + b_i \right)$$

**Cell State Update:** Combines the previous cell state and new candidate values $\tilde{c}^t$:

$$c^t = f^t \odot c^{t-1} + i^t \odot \tanh(W_x^c \tilde{x}^t + U_h^c h^{t-1} \cdot \gamma(\Delta t^t) + b_c)$$

**Output Gate:** Modulates the updated cell state $c^t$ to compute the next hidden state:

$$o^t = \sigma \left( W_x^o \tilde{x}^t + U_h^o h^{t-1} \cdot \gamma(\Delta \tilde{t}) + b_o \right)$$

The hidden state is then:

$$h^t = o^t \odot \tanh(c^t)$$

This setup ensures that for short intervals ($\Delta t < 1$), the model retains more of the past hidden state, while for longer intervals ($\Delta t > 1$), the model reduces the influence of $h^{t-1}$ and focuses on new information in $x^t$.

## 2.2 BI-DIRECTIONAL MECHANISM

In Electronic Health Records (EHRs), observations are often reliable for a certain period in a *bidirectional* manner. This is especially relevant in conditions like sepsis, where early symptoms may be subtle. Patients progressing into septic shock and those who do not may present with similar symptoms initially. A unidirectional LSTM processes only past observations and may mislabel these early states. In contrast, a bidirectional model re-evaluates these subtle symptoms by leveraging future observations, improving differentiation between shock and non-shock cases.

An analysis of the CCHS dataset supports this hypothesis. The results show that while early-stage distributions of shock and non-shock patients may appear similar, differences become clearer in later stages. The bidirectional approach helps detect subtle early-stage differences by incorporating future data, particularly as sepsis progresses.

### 2.2.1 PAST DECIDES THE FUTURE

In early sepsis, symptoms often resemble those of non-shock cases. The bidirectional model's **forward pass** captures these early signs and uses them to predict future states, modeling how past data influences future outcomes. The forward hidden state at time step $t$, $h_t^{\rightarrow}$, captures information up to $t$ and is used to predict future states: $h_t^{\rightarrow} = \text{LSTM}^{\rightarrow}(x^1, \ldots, x^t)$. While the forward pass captures past information, it is limited in adjusting predictions based on future observations, where the **backward pass** becomes crucial.

### 2.2.2 FUTURE ADJUSTS THE PAST

As sepsis progresses, the backward pass allows the model to re-interpret earlier observations using future data, which is particularly helpful for long trajectories like sepsis, where later symptoms help clarify earlier, more subtle signs. The backward hidden state $h_t^{\leftarrow}$ processes the sequence in reverse order, from $T$ to $t$: $h_t^{\leftarrow} = \text{LSTM}^{\leftarrow}(x^T, \ldots, x^t)$. When combined with the forward hidden state $h_t^{\rightarrow}$, the complete hidden state at time $t$ is $h_t = [h_t^{\rightarrow}; h_t^{\leftarrow}]$.

This joint representation allows the model to re-evaluate early symptoms based on more concrete symptoms that emerge later, providing a comprehensive understanding where future data can influence past interpretations.

### 2.3 BI-DIRECTIONAL MECHANISM WITH TIME EMBEDDING

The Bi-directional Time-Aware LSTM (Bi-T-LSTM) extends the conventional Bi-LSTM by incorporating time-awareness to handle irregularly spaced observations, which is common in Electronic Health Records (EHRs). This allows the hidden states to be updated based on the actual time intervals between consecutive events.

### 2.3.1 TIME-AWARE INPUT REPRESENTATION

Standard LSTMs assume regular intervals between events ($\Delta t = 1$), which is unrealistic in medical data. Time embeddings are introduced to capture the actual time difference $\Delta t$ between consecutive observations. For each time step $t$, the time difference is embedded as $e^t = E_t(\Delta t^t)$, and the augmented input becomes $\tilde{x}^t = \text{concat}(x^t, e^t)$, enabling the model to account for varying intervals.

### 2.3.2 FORWARD PASS

In the forward pass, the hidden state $h_t^{\rightarrow}$ is updated using $\tilde{x}^t$ and the previous hidden state $h_{t-1}^{\rightarrow}$, with the time decay function $\gamma(\Delta t) = \exp(-\alpha \Delta t)$ adjusting how much past information is retained:

$$h_t^{\rightarrow} = f^{\rightarrow}(\tilde{x}^t, h_{t-1}^{\rightarrow}), \quad f_t = \sigma(W_x^f \tilde{x}^t + U_h^f h_{t-1}^{\rightarrow} \cdot \gamma(\Delta t) + b_f)$$

This ensures that longer intervals cause faster decay of past information, while shorter intervals retain more of the hidden state.

### 2.3.3 BACKWARD PASS

In the backward pass, $h_t^{\leftarrow}$ is updated similarly but in reverse, processing the sequence from $T$ to $t$:

$$h_t^{\leftarrow} = f^{\leftarrow}(\tilde{x}^t, h_{t+1}^{\leftarrow}), \quad f_t = \sigma(W_x^f \tilde{x}^t + U_h^f h_{t+1}^{\leftarrow} \cdot \gamma(\Delta t) + b_f)$$

### 2.3.4 COMBINING FORWARD AND BACKWARD STATES

At each time step, the forward and backward hidden states are concatenated as $h_t = [h_t^{\rightarrow}; h_t^{\leftarrow}]$, allowing the model to utilize both past and future information, improving prediction accuracy for irregularly sampled time series such as EHRs.

## 3 EXPERIMENT

We have used three EHR datasets for this study. One we gathered from Christiana Care Health System Health System (CCHS) from July, 2013 to December, 2015, 2015. MIMIC-III, which is openly available data contained in MIMIC-II , which were collected between 2001 to 2008, and augments it with newly collected data between 2008 to 2012.Johnson et al. (2016) And Mayo from July, 2013 to December, 2015 (same date range as CCHS).

## 3.1 DATASET

Each dataset contains various status of each patient record's as with its unique visit identifier in its time sequence. From each sequence we define our study population with suspected sepsis infection, which is identified by the presence of any type of antibiotic, antiviral, or antifungal administration, or a positive test result of Point of Care Rapid (PCR).The definition of study population and the following data prepossessing were determined by leading clinicians from CCHS and Mayo Clinic. With these, we were able to identify 52,919 patient visits with suspected infection from CCHS dataset, 30,415 patient visits with suspected infection from MIMIC-III dataset, and 121,019 patient visits with suspected infection from Mayo dataset. From this, we conducted preprocessing as following:

**Missing data handling:** We handled missing data in both dataset by first forward-filling vitals, 6 sepsis progression-related feature, ('HeartRate', 'RespiratoryRate', 'PulseOx', 'SystolicBP', 'DiastolicBP', 'Temperature') for 8 hours and 9 lab values ('BandsUnits','BUN', 'Lactate', 'Platelet', 'Creatinine', 'BiliRubin', 'WBC', 'Procalcitonin', 'CReactiveProtein') for 24 hours. And Mean-fill the remaining missing values.

**Labeling for Septic Shock:** The most common method of clinical labeling relies on the International Classification of Diseases, Ninth Revision (ICD-9). While it serves a vital role in billing and administrative tasks, and to some extent in clinical documentation, it might not be ideally suited for detailed clinical analysis. One significant limitation is its lack of specificity in time-sensitive data; for instance, ICD-9 codes do not provide information about the exact timing of critical events such as the onset of septic shock. Vorwerk et al. (2009) Therefore, for more accurate identification of septic shock in our task, we have adopted the criteria set forth in the Third International Consensus Definitions for Sepsis and Septic Shock.Singer et al. (2016) Combining with the rule made by our clinicians identifing septic shock at each event as vasopressors, the presence of persistent hypotension (systolic blood pressure less than 90 mmHg or mean arterial pressure less than 65 mmHg for more than one hour), or a decrease in SBP of 40 mmHg or more within an eight-hour period.

**Sampling:** By applying both ICD-9 and the rule created by our clinicians, we have identified 1,869 shock-positive visits and 23,901 shock-negative visits within the CCHS dataset. Additionally, there are 2,459 shock-positive visits and 29,800 shock-negative visits from MIMIC-III dataset. And 3,499 shock-positive visits and 30,201 shock-negative visits within the Mayo dataset. Given the imbalance between positive and negative visits in both datasets, we conducted stratified random sampling on the shock-negative visits. This approach was taken to maintain the same underlying distribution of age, gender, ethnicity, length of stay, and the number of records in both the shock-positive and shock-negative groups. As a result of this process we were able to obtain the final dataset for CCHS with 3,738 visits (1,869 positive visits and 1,869 negative visits), MIMIC-III with 4,918 visits ( 2,459 positive visits and 2,459 negative visits), and Mayo with 6,998 visits (3,499 positive visits and 3,499 negative visits).

## 3.2 PREDICTION TASK

In our early prediction task, our objective is to forecast the development of septic shock in patients. For this, we utilize the patient's EHR up to n hours, early prediction window, prior to either the onset of septic shock or the end of the EHR sequence. Our methodology involves aligning the patient cases to the point of septic shock onset and the control cases to the end of their respective sequences; we will call this right-aligned. We then include all available EHR data up until n hours before these end

Figure 1: Event level early prediction (right aligned).

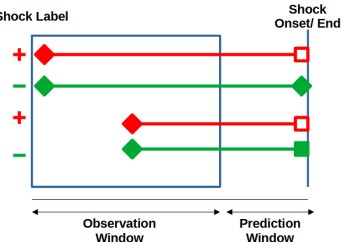

points. Essentially, our prediction model operates within an n-hour window leading up to the septic shock event or the (as shown in Figure.1) conclusion of the EHR sequence, aiming to accurately predict the likelihood of a patient developing septic shock in this time frame.

This task is challenging as the model must predict the occurrence of septic shock n-hours before it happens, based on all available data up to that point. This requires the model to identify subtle and possibly early indicators of septic shock that may not be as pronounced or clear as they would be closer to the event.

## 3.3 NESTED CROSS-VALIDATION WITH GRID SEARCH FOR HYPERPARAMETER TUNING

Nested Cross-Validation, coupled with a grid search approach, was employed to fine-tune hyper-parameters and evaluate the model's performance. This method is crucial to ensure the robustness and generalizability of our models across different datasets. In our study, we implemented three base types of neural network models: Long Short-Term Memory (LSTM), Variational Recurrent Neural Networks (VRNN), and Transformer. Building upon these, we introduced six variations by incorporating two key mechanisms: Bidirectional and Time-Aware models.

## 3.4 PERFORMANCE METRICS

To comprehensively evaluate the performance of our neural network models, we have selected a range of metrics, each offering unique insights into different aspects of model performance. These include accuracy, recall, precision, F1 score, and Area Under the Curve (AUC). Special emphasis is placed on the AUC due to its significance in our study. Our primary focus is on the AUC, as it serves as a robust indicator of the model's discriminative power between septic shock and non-septic shock cases, making it the most relevant metric for evaluating early prediction models in this context.

## 4 RESULTS

Our comprehensive evaluation of various neural network models for early prediction of septic shock yielded insightful findings, with each model demonstrating unique performance characteristics across different early prediction windows (4 to 32).

Table 1: F1 and AUC scores of selected models evaluated on MIMIC, CCHS, and Mayo for the 4-32 hours early prediction window.

| Test Domain | Model | $F_1$ Score | AUC |
|---|---|---|---|
| CCHS | VRNN | 0.857($\pm$0.017) | 0.8433($\pm$0.015) |
| | LSTM | **0.8624**($\pm$0.012) | **0.8487**($\pm$0.028) |
| | Transformer | **0.8635**\*\*($\pm$0.010) | **0.8645**\*\*($\pm$0.021) |
| | RAPT | 0.8738($\pm$0.011) | 0.889($\pm$0.015) |
| | Bi-T-VRNN | 0.8767($\pm$0.021) | 0.8945($\pm$0.010) |
| | Bi-T-LSTM | **0.881**\*\*($\pm$0.007) | **0.9017**\*\*($\pm$0.012) |
| | Bi-T-Transformer | **0.8796**($\pm$0.019) | **0.8976**($\pm$0.013) |
| Mayo | VRNN | 0.8822($\pm$0.014) | 0.8643($\pm$0.017) |
| | LSTM | **0.8834**($\pm$0.018) | **0.8696**($\pm$0.011) |
| | Transformer | **0.8886**\*\*($\pm$0.021) | **0.8712**\*\*($\pm$0.017) |
| | RAPT | 0.8899($\pm$0.024) | **0.8893**($\pm$0.018) |
| | Bi-T-VRNN | **0.8939**\*\*($\pm$0.019) | 0.8887($\pm$0.018) |
| | Bi-T-LSTM | **0.8931**($\pm$0.012) | **0.8907**\*\*($\pm$0.021) |
| | Bi-T-Transformer | 0.8926($\pm$0.014) | 0.8853($\pm$0.010) |
| MIMIC | VRNN | **0.8562**($\pm$0.013) | 0.878($\pm$0.023) |
| | LSTM | 0.8556($\pm$0.006) | **0.8823**($\pm$0.015) |
| | Transformer | **0.8567**\*\*($\pm$0.026) | **0.8824**\*\*($\pm$0.011) |
| | RAPT | **0.8611**($\pm$0.015) | 0.8867($\pm$0.021) |
| | Bi-T-VRNN | 0.8606($\pm$0.025) | 0.8852($\pm$0.012) |
| | Bi-T-LSTM | **0.8615**\*\*($\pm$0.013) | **0.8902**\*\*($\pm$0.027) |
| | Bi-T-Transformer | 0.8591($\pm$0.019) | **0.8881**($\pm$0.017) |

· The *best* and the *second best* models are labeled with \*\* after the number and bolded for emphasis.

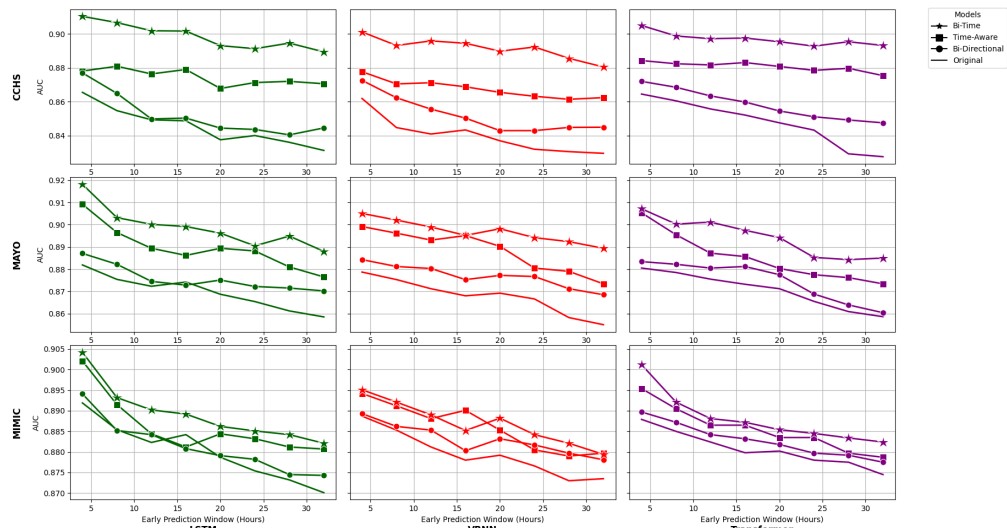

Figure 2: AUC for each model based on the early prediction window on three datasets

Table 1 presents the experimental results for an early prediction window (4 to 32 hours) across the CCHS, MIMIC-III, and Mayo datasets. Among the original models such as VRNN, LSTM, and Transformer, the Transformer consistently achieved the best performance in terms of F1 score and AUC. However, when we integrated the two mechanisms, bidirectionality and time-awareness, into each model, their performance improved significantly. Notably, with these enhancements, the LSTM model outperformed all other models across all three datasets. the introduction of bidirectional and time-aware mechanisms significantly enhances their ability to handle the irregularities and directional dependencies present in EHR data. We have also included the Pre-training of Time-Aware Transformer (RAPT)(Ren et al., 2021) as a baseline to further demonstrate the robustness of combination of bidirectioanlity and time-aware mechanisms. As seen in Table 1, RAPT performs consistently well, achieving competitive F1 scores and AUC values across all three datasets. However, when compared to the bidirectional and time-aware models, RAPT's performance, while strong, is slightly outperformed, particularly by the Bi-T-LSTM model.

Figure 2 presents the AUC results for each model across various early prediction windows using the CCHS, MIMIC, and Mayo datasets. Each row corresponds to a dataset, while the columns represent the three models: LSTM, VRNN, and Transformer. The models are evaluated with four mechanisms: Bi-Time, Time-Aware, Bi-Directional, and the Original versions. Across all datasets and models, the Bi-Time mechanism consistently outperforms the others in terms of AUC, particularly as the prediction window increases. This demonstrates that incorporating both bidirectionality and time-awareness significantly improves predictive performance for early septic shock detection, regardless of the model type. While Bi-T-LSTM often achieves the highest AUC values, VRNN and Transformer models also benefit significantly from these mechanisms. This highlights the broad applicability of bidirectionality and time-awareness, as they consistently enhance the predictive capabilities of all tested models.

Figure 3: Critical difference diagram for AUC on the Mayo Dataset

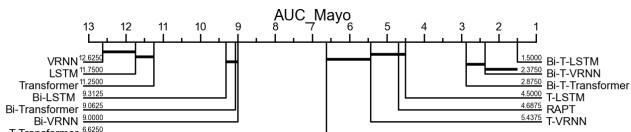

Figures 3 provide a Critical Difference (CD) diagram Ismail Fawaz et al. (2019) representing the statistical significance of the differences in AUC scores between models for the Mayo datasets, respectively; other two cd-diagrams can be found in the appendix. The CD diagrams, constructed using the Wilcoxon signed-rank test at an alpha level of 0.05, illustrate the relative ranking of model performance.

In Figure 3, the critical difference diagram clearly illustrates that bi-directional time-aware models consistently outperform other models in terms of AUC across various early prediction windows. The integration of bi-directional processing and time-awareness is essential for capturing complex temporal dependencies and improving predictive accuracy. Notably, Bi-T-LSTM ranks the highest among all models, further reinforcing its ability to effectively leverage these mechanisms to achieve superior performance.

## 5 RELATED WORK

While previous research has incorporated one or two of these mechanisms into some models, no prior work, to our knowledge, has comprehensively evaluated all three mechanisms across the foundational models of Long Short-Term Memory (LSTM), Variational Recurrent Neural Networks (VRNN), and Transformers. Although both Bidirectionality and Time-awareness have been individually applied to LSTMs, this study is the first to combine both mechanisms within the LSTM framework and demonstrate the significant performance improvements that result. Additionally, while VRNNs have been shown to outperform LSTMs on certain Electronic Health Records (EHR) datasets (Zhang et al., 2017a; Khoshnevisan & Chi, 2020), no prior research has explored the integration of Bidirectionality or Time-awareness with VRNNs.

**Recurrent Neural Network (RNN) & LSTM:** The most popular deep learning framework adaptive to time-series EHR prediction is Recurrent Neural Network (RNN) due to its capability of handling long-range temporal dependencies. Popular RNN variants are the long short-term memory (LSTM) and gated recurrent unit (GRU) models. Lipton et al. Lipton et al. (2015) were the first to apply LSTM networks for multi-label prediction in EHR data from ICU patients. The promising results from this work have opened up a line of research around variations of RNN by addressing various challenges existing in EHR. Che et al. proposed a variation of the recurrent GRU cell (GRU-D) which attempts at better handling of missing values in clinical time-series Che et al. (2018). Their GRU-D networks show improved AUC on two ICD-9 classification and mortality prediction tasks. DeepCare introduces time parameterizations to enable irregular timing by moderating the forgetting dynamics in LSTM Pham et al. (2016). Also, ATTAIN is a time-aware LSTM model that models the inherent irregular time intervals in EHR data by defining a decay function correlated to all previous time steps Zhang (2019). Combining Convolutional Neural Networks (CNN) with LSTM have also been explored for septic shock early prediction Lin et al. (2018). Furthermore, this study has shown that combining static information, such as demographics, and dynamic information of EHRs can be effective for accurate clinical event prediction. In a similar study, Esteban et al. Esteban et al. (2016) used deep models for predicting the onset of complications relating to kidney transplantation. They combined static and dynamic features as input to various types of RNNs. The results demonstrated that the GRU-based network in conjunction with static patient data outperformed other deep variants. Moreover, RNN or LSTM with attention networks is widely developed to improve the interpretability of such models in the medical domain. As a pioneer work, RETAIN Choi et al. (2016b) applied a two-level attention mechanism to identify meaningful visits and specific features that contribute to the prediction. Similarly, Dipole Ma et al. (2017) employs an attention-based bidirectional RNN for diagnosis prediction task.

**Variational Recurrent Neural Network (VRNN)** Generative models with recurrent structures, such as Variational Recurrent Neural Networks (VRNN), were first introduced by Chung et al. in 2015 Chung et al. (2015). Compared to conventional generative models, Variational Auto-encoder (VAE) and VRNN can model more complex conditional distributions and variability in temporal progression, hence representing more complex patterns that can potentially improve performance for different prediction tasks. Recurrent variational models have shown success in different fields including speech modeling Lee et al. (2018); Chien et al. (2017), natural language processing Pineau & de Lara (2019), object tracking in video Hoy et al. (2018), and recommender systems Christodoulou et al. (2017). Moreover, the generative power of such models can compensate for the high missing rate in input data. However, applications of VRNN in the medical domain, especially for multivariate healthcare time-series data that is associated with a high missing rate are surprisingly underexplored. In a study, Zhang et al. Zhang et al. (2017a) proposed an end-to-end architecture that employs a VRNN for learning robust and generalizable features from lab test data. This model is simultaneously trained with a neural network (NN) to learn diagnosis decision-making. The results show that VRNN+NN significantly outperforms other deep learning models while offering a good imputation for missing values in EHR. In another study, Zhang et al. demonstrate the superiority of

the VRNN model for missing data imputation in EHR data, by showing its impact on the improvement of the septic shock early prediction performance Zhang et al. (2019b). VRADA Purushotham et al. (2016), Variational Recurrent Adversarial Deep Domain Adaptation, is a VRNN-based domain adaptation framework that is trained adversarially to capture complex temporal relationships that are domain-invariant. Experiments on real-world EHR data have demonstrated that learning temporal dependencies using VRNN improves VARDA's ability to create domain-invariant representations, and results in outperforming state-of-the-art domain adaptation approaches. Thus, in this study, we leverage VRNN architecture to capture complex temporal dependencies in EHR data, while handling missing values in such data.

**Attention mechanisms** In recent years, attention mechanisms are extensively explored to interpret the model output and greatly improve the prediction performance. For example, RETAIN applies a reverse time attention mechanism in an RNN Choi et al. (2016b) and Dipole Ma et al. (2017) uses the similar attention networks for diagnosis prediction. Another challenge associated with EHR data, time irregularity, has also been tackled. T-LSTM Baytas et al. (2017) divides short-term from the previous cell memory, and adjusts it with a time-aware mechanism. In Pham et al. (2016), the time intervals are used to modify the forget gate of LSTM. In Che et al. (2018), time gaps are made regular through data imputation methods. Finally, Health-ATM Ma et al. (2018) extracts patient information patterns with attentive and time-aware models through RNN and Convolutional Neural Networks (CNN). Compared with the prior works, our proposed method explores different attention mechanisms to generate weights for the past events while handling the time irregularity in EHRs. For acute medical conditions such as septic shock, it is extremely significant to identify critical and timely meaningful events.

**Transformer models:** Transformer models, initially introduced by Vaswani et al. Vaswani et al. (2017), have revolutionized sequence modeling through their use of self-attention mechanisms, which allow for capturing dependencies regardless of their distance in the sequence. Unlike RNN-based models, Transformers can process input sequences in parallel, leading to significant improvements in computational efficiency. In the context of EHRs, Transformers can handle the complex and irregular structure of medical data more effectively. For instance, models such as BERT Devlin et al. (2018), which is based on the Transformer architecture, are inherently bidirectional, enabling them to consider both past and future contexts simultaneously. Similarly, building upon Transformer architecture, RAPT (Pre-training of Time-Aware Transformer) Ren et al. (2021) incorporates time-awareness, making it particularly well-suited for clinical time series data where measurements are irregularly sampled. This capability is particularly advantageous for clinical event prediction and patient outcome modeling. Recent studies have demonstrated the potential of Transformer models in medical applications, including EHR data for tasks such as disease prediction Li et al. (2020), mortality risk assessment Huang et al. (2019), and temporal phenotyping Rasmy et al. (2021). By leveraging the self-attention mechanism, Transformer models can capture intricate relationships within patient data, making them a powerful tool for healthcare analytics.

## 6  CONCLUSION

Our comprehensive evaluation of neural network architectures provides critical insights into the predictive modeling of septic shock, particularly highlighting the power of bidirectionality and time-awareness in improving performance across multiple models. While LSTMs consistently demonstrated superior performance when integrated with both mechanisms, our findings confirm that the impact of bidirectionality and time-awareness extends beyond individual models, enhancing VRNNs and Transformers as well.

In conclusion, this study highlights the importance of bidirectionality and time-awareness in disease progression modeling. These two mechanisms enable LSTMs to excel in capturing complex temporal relationships, but they also significantly improve the performance of VRNNs and Transformers. This underscores the potential of these mechanisms to set a new benchmark for early detection of critical conditions like septic shock, pushing the boundaries of model performance across different architectures.

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

## A  INCORPORATING TIME INTERVALS ($\Delta t$) IN VRNN

In the context of VRNN, the time intervals $\Delta t$ between consecutive observations play a crucial role in accurately modeling the temporal dynamics and latent structure of clinical data. Similar to the LSTM, a standard VRNN assumes uniform time intervals ($\Delta t = 1$) between consecutive events, which can limit its ability to handle irregular time series data, such as electronic health records (EHR), where time intervals between observations can vary significantly.

In the case of VRNN, the model combines a recurrent neural network (RNN) with a variational autoencoder (VAE) to capture both sequential dependencies and stochastic latent dynamics. To effectively handle irregularly sampled time series, the model must incorporate the varying time intervals $\Delta t$, allowing it to adjust its latent variable dynamics and memory mechanisms based on the actual time between events.

### A.1 TIME-AWARE INPUT REPRESENTATION

To incorporate $\Delta t$ into the VRNN, we first augment the input at each time step with a time embedding. Let $\Delta \tilde{t}$ represent the time difference between consecutive observations. The time embedding $e$ is generated using a time embedding matrix $E$ as $e = E(\Delta \tilde{t})$. The time embedding $e$ is concatenated with the input features $x$, resulting in an augmented input $\tilde{x} = \text{concat}(x, e)$. This augmented input $\tilde{x}$ is then passed through the VRNN, ensuring that the model is aware of the actual time intervals between observations and can adjust the latent variable dynamics accordingly.

### A.2 ENCODER WITH TIME INTERVALS

In the VRNN, the encoder is responsible for inferring the latent variable $z$ based on the augmented input $\tilde{x}$ and the previous hidden state $h$. To incorporate time intervals, the encoder is modified to include the time-aware augmented input $\tilde{x}$, which allows the inference of $z$ to reflect the time elapsed between consecutive observations. The encoder can be expressed as: $q(z|\tilde{x}, h) = \mathcal{N}(\mu_z, \sigma_z)$. Here, the time-aware augmented input $\tilde{x}$ helps the model infer latent variables $z$ that are sensitive to the time interval $\Delta \tilde{t}$.

### A.3 PRIOR MODEL WITH TIME INTERVALS

The prior distribution in VRNN is defined to regularize the latent space by learning a prior distribution over the latent variable $z$ conditioned on the previous hidden state $h$. In a time-aware VRNN, this prior distribution also incorporates the time interval between consecutive observations. The prior model can be written as: $p(z|h) = \mathcal{N}(\mu_p, \sigma_p)$. To account for irregular time intervals, the hidden state $h$ is updated based on the time-aware augmented input $\tilde{x}$ and the latent variable $z$, ensuring that the latent variable reflects both the observed data and the temporal dynamics.

### A.4 DECODER WITH TIME INTERVALS

The decoder in VRNN is responsible for reconstructing the observed input $x$ from the latent variable $z$ and the previous hidden state $h$. In a time-aware VRNN, the reconstruction process incorporates the time interval $\Delta \tilde{t}$ by decoding the augmented input $\tilde{x}$, which includes the time embedding $e$. The generative model can be expressed as: $p(x|z, h) = \mathcal{N}(\mu_x, \sigma_x)$. By incorporating the time-aware augmented input $\tilde{x}$, the model can generate realistic reconstructions that account for the varying time intervals between observations.

### A.5 RECURRENT UPDATE WITH TIME DECAY

In the time-aware VRNN, the hidden state $h$ is updated based on both the augmented input $\tilde{x}$ and the latent variable $z$. Similar to the LSTM, we introduce a time decay function $\gamma(\Delta \tilde{t})$ to modulate the influence of the previous hidden state $h$ based on the time interval $\Delta \tilde{t}$. This decay function can be defined as: $\gamma(\Delta \tilde{t}) = \exp(-\alpha \Delta \tilde{t})$, where $\alpha$ is a learnable parameter controlling the decay rate. The recurrent update in VRNN is then adjusted as: $h = \text{GRU}(\tilde{x}, z, h \cdot \gamma(\Delta \tilde{t}))$. This modification allows the VRNN to adjust the influence of the past hidden state based on the time elapsed between events. For short intervals ($\Delta \tilde{t} < 1$), the decay function slows down, retaining more of the past hidden state $h$, while for long intervals ($\Delta \tilde{t} > 1$), the decay function speeds up, reducing the influence of the past and allowing the model to focus on the new information in $x$.

In this formulation, the time embedding ensure that the model captures the irregular time intervals between observations, allowing for better latent variable inference and more accurate predictions.

## B INCORPORATING TIME INTERVALS ($\Delta t$) IN TRASFORMERR

Transformers rely on self-attention mechanisms to model relationships between different elements of a sequence. In standard Transformers, positional encodings are used to inject information about the order of the sequence, as the attention mechanism itself is invariant to the position of tokens. However, these positional encodings typically assume that time intervals between observations are uniform. In the case of irregular time series data, such as in electronic health records (EHR), time

intervals $\Delta t$ between observations vary significantly. To address this, we modify the Transformer to incorporate time intervals $\Delta \tilde{t}$.

## B.1   TIME-AWARE INPUT REPRESENTATION

In a time-aware Transformer, the time intervals $\Delta \tilde{t}$ between consecutive observations must be embedded alongside the input features. Let $x_i$ represent the input features at time step $i$, and let $\Delta \tilde{t}_i$ represent the time interval between observations $i - 1$ and $i$. We introduce a time embedding $e_i$ to capture this time interval: $e_i = E(\Delta \tilde{t}_i)$. The time embedding $e_i$ is concatenated with the input features $x_i$, resulting in an augmented input $\tilde{x}_i = \text{concat}(x_i, e_i)$. The augmented input $\tilde{x}_i$ is then passed through the Transformer's attention mechanism, enabling the model to account for irregular time intervals during the self-attention computation.

## B.2   SELF-ATTENTION WITH TIME INTERVALS

The core of the Transformer is its self-attention mechanism, which computes pairwise attention scores between different positions in the input sequence. The attention weights between time steps $i$ and $j$ are computed using a dot product between their query and key vectors. In a standard Transformer, the position of each time step is encoded via positional embeddings. However, to capture the irregularity in time intervals, we adjust the attention mechanism to consider the time intervals $\Delta \tilde{t}$. Given query vector $q_i$, key vector $k_j$, and the time embedding $e_i$ and $e_j$ for time steps $i$ and $j$, we modify the attention scores as:

$$\text{Attention}(q_i, k_j) = \frac{(q_i + e_i)(k_j + e_j)^T}{\sqrt{d_k}}$$

Here, the time embeddings $e_i$ and $e_j$ are added to the query and key vectors, ensuring that the attention mechanism is aware of the time intervals between the observations. The superscript $^T$ indicates the transpose of the key vector $(k_j + e_j)$, which allows the dot product to be computed properly with the query vector $(q_i + e_i)$. This adjustment allows the Transformer to weight observations based on both their feature similarity and the time elapsed between them.

## B.3   POSITION AND TIME EMBEDDINGS

Standard Transformers use positional encodings to inject information about the order of the sequence. In a time-aware Transformer, we combine positional encodings with time embeddings to capture both the relative positions of the observations and the actual time intervals between them. Let $\text{PosEnc}(i)$ denote the positional encoding for time step $i$, and let $e_i = E(\Delta \tilde{t}_i)$ represent the time embedding for the time interval $\Delta \tilde{t}_i$. The final embedding for each input is the sum of these two components: $\text{FinalEmbedding}_i = \tilde{x}_i + \text{PosEnc}(i)$. This embedding is passed through the Transformer layers, where the attention mechanism processes the inputs based on both their positional information and the time intervals between them.

## B.4   TIME DECAY IN ATTENTION SCORES

To further emphasize the importance of time intervals, we introduce a time decay function $\gamma(\Delta \tilde{t})$ that modulates the attention scores based on the time elapsed between observations. The time decay function is defined as: $\gamma(\Delta \tilde{t}) = \exp(-\alpha \Delta \tilde{t})$, where $\alpha$ is a learnable parameter controlling the decay rate. This decay function is applied to the attention weights, reducing the influence of distant observations in time.

$$\text{AttentionWeight}_{i,j} = \frac{\gamma(\Delta \tilde{t}) \cdot (q_i + e_i)(k_j + e_j)^T}{\sqrt{d_k}}$$

Incorporating time decay into the attention mechanism further enhances the model's ability to weigh observations appropriately. Observations that occur closer together in time are given more attention, while those separated by larger time intervals are weighted less. This allows the model to retain long-term dependencies when needed while focusing more on recent observations for short intervals.

## C    BI-DIRECTIONAL VARIATIONAL RECURRENT NEURAL NETWORK (BI-VRNN)

In a similar manner to LSTMs, a Bi-directional VRNN is capable of capturing both past and future contexts in sequential data. The VRNN extends traditional RNNs by incorporating a latent variable at each time step, which allows for better modeling of complex time series data like EHRs. When this model is extended in a bidirectional way, it leverages future observations to re-interpret earlier subtle symptoms, which is particularly useful in sepsis shock prediction.

The Bi-VRNN uses both a forward and backward pass, similar to a Bi-LSTM, but incorporates a latent variable $z^t$ at each time step to model the stochastic dynamics of the sequence. The hidden state in both directions is updated based on the current input and latent variable.

### FORWARD PASS

In the forward pass, the hidden state $h_t^{\rightarrow}$ is updated at each time step $t$ using the input $x^t$ and the previous hidden state $h_{t-1}^{\rightarrow}$, along with a latent variable $z^t$: $h_t^{\rightarrow} = f^{\rightarrow}(x^t, h_{t-1}^{\rightarrow}, z^t)$. At each time step, the latent variable $z^t$ is inferred using the recognition model: $q(z^t \mid x^t, h_{t-1}^{\rightarrow}) \sim \mathcal{N}(\mu_t^{\rightarrow}, \sigma_t^{\rightarrow})$. The generative model for reconstructing the input $x^t$ from the latent variable $z^t$ and the hidden state $h_t^{\rightarrow}$ is given by: $p(x^t \mid z^t, h_t^{\rightarrow}) \sim \mathcal{N}(\mu_x^t, \sigma_x^t)$.

### C.1    BACKWARD PASS

Similarly, in the backward pass, the hidden state $h_t^{\leftarrow}$ is updated by processing the sequence in reverse (from time step $T$ to $t$): $h_t^{\leftarrow} = f^{\leftarrow}(x^t, h_{t+1}^{\leftarrow}, z^t)$. Here, $h_t^{\leftarrow}$ is computed using the input $x^t$, the next hidden state $h_{t+1}^{\leftarrow}$, and the latent variable $z^t$ inferred from future information.

### C.2    COMBINING FORWARD AND BACKWARD STATES

At each time step $t$, the forward and backward hidden states are concatenated to form the final hidden state $h_t$: $h_t = [h_t^{\rightarrow}; h_t^{\leftarrow}]$. This allows the model to utilize both past and future context in making predictions.

## D    BI-DIRECTIONAL TRANSFORMER

Transformers are highly effective in modeling long-range dependencies in sequential data due to their self-attention mechanisms. In a Bi-directional Transformer, the self-attention mechanism is adapted to include time embeddings that capture both the sequence order and the time intervals between observations, allowing for a more nuanced understanding of the patient's health trajectory over time. Similar to BERT(Devlin et al., 2018), a Bi-directional Transformer leverages the self-attention mechanism to capture dependencies between all time steps in a sequence. In a bidirectional setup, it incorporates both forward and backward attention to model how past and future observations relate to each other.

### D.1    FORWARD SELF-ATTENTION

In the forward pass, the Transformer uses self-attention to compute the relationships between the input at time $t$ and all previous inputs. At each time step $t$, the input sequence $x^t$ is projected into query $q^t$, key $k^t$, and value $v^t$ vectors, which are learned transformations of the input features:

$$q^t = W_q x^t, \quad k^t = W_k x^t, \quad v^t = W_v x^t$$

The self-attention mechanism computes how much focus the model should place on each previous time step's input by calculating the attention score between the current query $q^t$ and all previous keys $k^{1:t}$. This attention score is used to weigh the corresponding value vectors $v^{1:t}$, effectively allowing the model to focus on the most relevant parts of the input sequence. The attention mechanism is defined as:

$$\text{Attention}(q^t, k^{1:t}) = \text{softmax}\left(\frac{q^t (k^{1:t})^\top}{\sqrt{d_k}}\right) v^{1:t}$$

Here, the dot product between $q^t$ and $k^{1:t}$ captures the similarity between the current time step and previous time steps, and the division by $\sqrt{d_k}$ helps prevent overly large values that could dominate the softmax computation. The result is a weighted sum of the value vectors, where more relevant time steps (based on their attention scores) contribute more to the final output at time step $t$. This mechanism enables the model to dynamically attend to important past information, making it well-suited for handling dependencies across time in sequential data.

### D.2 BACKWARD SELF-ATTENTION

In the backward pass, the model similarly computes the attention between the current time step $t$ and all future time steps. The backward attention is computed as:

$$\text{Attention}(q^t, k^{t:T}) = \text{softmax}\left(\frac{q^t (k^{t:T})^\top}{\sqrt{d_k}}\right) v^{t:T}$$

### D.3 COMBINING FORWARD AND BACKWARD ATTENTION

Finally, the model combines the results from the forward and backward attention mechanisms to compute the final representation for each time step:

$$h^t = [h_t^{\rightarrow}; h_t^{\leftarrow}]$$

Here, $h_t^{\rightarrow}$ represents the forward context and $h_t^{\leftarrow}$ represents the backward context, combining information from both past and future time steps.

## E ADDITIONAL INFORMATION ON EXPERIMENT SETUP FOR EACH MODEL

We describe the setup of our experiments designed to evaluate the efficacy of various neural network architectures in predicting septic shock. We have developed and tested twelve different models, each with unique characteristics and approaches to handling temporal data in EHRs. All models were trained using the Adam optimizer with a learning rate of 0.001. These models include:

• **VRNN (Variational Recurrent Neural Network)**: The VRNN model features an encoder and a decoder. The encoder, a sequential neural network, maps the hidden state to a latent space represented by mean (mu) and log-variance (logvar). The decoder then reconstructs the input data from this latent representation, capturing the underlying patterns in the EHR data. At the core of the VRNN is an RNN layer nn.GRU with Pytorch, which processes the input data across time. This layer is essential for capturing the temporal dependencies present in the sequential data of EHRs. Then nn.Linear maps the output the decoder to the predicting septic shock. The model employs the reparameterization trick Kingma & Welling (2013) for the latent variables, enabling it to sample efficiently from the latent space during training.

• **Bi-VRNN (Bidirectional VRNN)**: Building upon the standard VRNN, the Bi-VRNN also features an encoder and a decoder for transforming hidden states into a latent space and reconstructing the input data, respectively. It utilizes the same reparameterization trick for handling latent variables. The key distinction lies in its bidirectional processing of sequential data, allowing the model to capture dependencies influenced by both preceding and subsequent events in the sequence.

• **T-VRNN (Time-aware VRNN)**: T-VRNN is an advanced neural network that combines the principles of variational autoencoders with time-aware recurrent neural networks. Using VRNN as its baseline, the T-VRNN maintains a similar structure, including the use of the reparameterization trick. A key unique feature of this model is its incorporation of time-aware encoding through a specialized time embedding layer. This layer translates time indices into a meaningful representation, which is then seamlessly integrated with traditional input features. This integration equips the model

with heightened sensitivity to the timing and sequence of events within EHR data. The time-aware encoding specifically enhances the model's capability to process and interpret sequences with irregular or significant time intervals between data points, a common characteristic in EHR datasets. By doing so, the T-VRNN becomes particularly adept at understanding and predicting outcomes in scenarios where temporal dynamics play a crucial role, making it exceptionally suitable for complex healthcare data analysis.

- **Bi-T-VRNN (Bidirectional Time-aware VRNN)**: Expanding on the T-VRNN's capabilities, the Bi-T-VRNN introduces bidirectional processing in its recurrent neural network layer. This crucial enhancement enables the model to analyze temporal sequences in both forward and backward directions, thus capturing a more comprehensive temporal context within EHR data. Like the T-VRNN, it employs a time embedding layer to convert time indices into an informative representation, further enriching the model's sensitivity to the timing of events. The Bi-T-VRNN's bidirectional architecture, combined with time-aware processing, significantly boosts its effectiveness in complex predictive task like predicting septic shock.

- **LSTM (Long Short-Term Memory)**: Our LSTM model uses PyTorch's nn.Module. It consists of a single-layer LSTM and a fully connected output layer. the LSTM layer processes sequences in a batch-first manner and includes a dropout of 0.2 for regularization.s

- **Bi-LSTM (Bidirectional LSTM)**: Our Bi-LSTM model uses PyTorch's nn.Module. The core LSTM layer is set up where it processes data in both forward and backward directions. This allows the model to capture dependencies from both past and future contexts in the sequence.

- **T-LSTM (Time-aware LSTM)**: Our T-LSTM model is a specialized version of LSTM designed to account for time intervals in the data. It extends PyTorch's nn.Module and includes an LSTM layer and a linear output layer. Unlike a regular LSTM, this model incorporates a 'TimeStep' feature to account for varying time intervals between observations in the EHR data.the model first concatenates the 'TimeStep' feature with the input data. It then initializes hidden and cell states and processes the input through the LSTM. The final predictions are based on the last hidden states, capturing both the sequential nature of the data and the time intervals between observations. The incorporation of the 'TimeStep' feature makes the T-LSTM model uniquely suited for our task. It allows the model to account for the timing of events, which is critical in predicting septic shock where the timing and sequence of medical events can provide key insights.

- **Bi-T-LSTM (Bidirectional Time-aware LSTM)**: The Bi-T-LSTM model extends the capabilities of a standard T-LSTM by incorporating bidirectional processing. This bidirectionality allows the model to capture temporal dynamics in both past and future directions. In the context of EHR data, this means the model can integrate information from both earlier and later stages of a patient's medical history, providing a more comprehensive analysis than a non-bidirectional approach.

- **Transformer**: The Transformer model Vaswani et al. (2017) is a neural network architecture designed for handling sequential data without relying on recurrent layers. Instead, it uses self-attention mechanisms to process the entire sequence of data simultaneously. This allows the model to capture long-range dependencies more effectively. Our implementation uses PyTorch's `nn.Transformer` module, consisting of multiple encoder layers that process the input data to predict septic shock. The model is particularly suited for handling sequences where capturing global context is essential.

- **Bi-Transformer (Bidirectional Transformer)**: The Bi-Transformer extends the standard Transformer architecture by incorporating bidirectional processing within its self-attention mechanisms. This enhancement allows the model to consider context from both preceding and succeeding events in the sequence, providing a more comprehensive understanding of the temporal dependencies in EHR data. This bidirectional approach enhances the model's ability to capture the full scope of the patient's medical history, leading to improved predictions.

- **T-Transformer (Time-aware Transformer)**: The T-Transformer uses PyTorch's `nn.Transformer` modules, consisting of multiple encoder and decoder layers. The input sequence is normalized, converted to tensors, and passed through the Transformer layers, with padding masks created to handle variable sequence lengths. This modification enables the model to account for the temporal aspect of data explicitly. It includes a specialized time embedding layer that translates time indices into a meaningful representation, which is then integrated with

the traditional input features. This time-aware encoding allows the model to process and interpret sequences with irregular or significant time intervals between data points, which is crucial for accurate predictions in medical applications such as early septic shock prediction.

• **Bi-T-Transformer (Bidirectional Time-aware Transformer)**: The Bi-T-Transformer combines the principles of bidirectional processing and time-aware encoding in the Transformer architecture. This model processes temporal sequences in both forward and backward directions, capturing a comprehensive temporal context within EHR data. The time embedding layer converts time indices into informative representations, which are integrated with the input features. The bidirectional and time-aware capabilities enable the Bi-T-Transformer to provide more accurate and insightful predictions by fully leveraging the timing and sequence of events in the data.

## F ADDITIONAL INFORMATION NESTED CROSS-VALIDATION WITH GRID SEARCH FOR HYPERPARAMETER TUNING

**Grid Search for Hyperparameter Optimization**: Grid Search Methodology: This approach systematically works through multiple combinations of parameter options, determined by a predefined 'grid' of hyperparameters. We conducted a grid search over several key parameters: learning rates (0.0001, 0.001, 0.01), batch sizes (64, 32,16,8), hidden dimensions (512,256,128, 64). And for VRNN based models Latent Dimensions (32,64,128). The grid search iterates through each combination of these parameters to determine which set produces the best model performance, typically assessed via a validation metric AUC.

**Outer Loop - Model Evaluation (2-fold CV)**: The dataset is divided into two distinct folds. In each iteration, one fold is for training (further divided in the inner loop for the grid search) and the other for testing. The test fold remains untouched during the training and hyperparameter tuning to avoid data leakage and ensure an unbiased evaluation. After training with the best hyperparameters, performance metrics (AUC) are calculated on this test fold.

**Inner Loop - Hyperparameter Tuning via Grid Search (10-fold CV)**: Within each training iteration of the outer loop, we perform a 10-fold cross-validation as part of the grid search. For each parameter combination, the model is trained on 9 folds and validated on the remaining fold. This is repeated for all folds and all parameter combinations. The average performance across these folds is computed for each set of parameters. The combination yielding the best average performance is selected as the optimal one for that training iteration.

**Reporting Results**: During the inner loop, we report the performance metrics like AUC and F1 score for the validation sets. This guides us in choosing the best hyperparameters (Validation performance). In the outer loop, these metrics are calculated for the test set, providing an assessment of the model's performance on unseen data (Test performance). This approach, incorporating both Nested Cross-Validation and Grid Search, ensures thorough hyperparameter optimization while maintaining an unbiased estimate of model performance. The grid search allows us to explore a range of parameter configurations systematically, while the nested cross-validation structure ensures that the model's evaluation is robust and generalizable to new data.

## G ADDITIONAL RESULTS FOR THREE DATASET AND OTHER DETAILS

In this section we provide additional details that are supplement to the Result section.

### G.1 DETAILED RESULTS

Table 2 presents the detailed experimental outcomes for a specific early prediction window (n=28) applied to the CCHS, MIMIC-III, and Mayo datasets. The focus of our hyperparameter tuning was to optimize the models for the best AUC metric. Across all datasets, models incorporating both bidirectionality and time-awareness consistently outperform their original counterparts, as reflected in both F1 scores and AUC values.

Across all datasets, Bi-T-Transformer and Bi-T-LSTM consistently emerged as top performers, particularly with respect to the AUC metric at a early prediction window (n = 28), confirming the ef-

Table 2: Performance comparison of different models across datasets at window 28.

| Test Domain | | Model | Accuracy | Precision | Recall | $F_1$ Score | AUC |
|---|---|---|---|---|---|---|---|
| CCHS | 1. | LSTM | 0.8324($\pm$0.015) | 0.8744($\pm$0.022) | 0.8771($\pm$0.018) | 0.8757($\pm$0.026) | 0.836($\pm$0.024) |
| | 2. | VRNN | 0.8412($\pm$0.017) | 0.8755($\pm$0.014) | 0.8793($\pm$0.013) | 0.8774($\pm$0.016) | 0.8305($\pm$0.018) |
| | 3. | Transformer | 0.8364($\pm$0.012) | 0.862($\pm$0.009) | 0.8664($\pm$0.011) | 0.8631($\pm$0.01) | 0.8292($\pm$0.012) |
| | 4. | RAPT | 0.8536($\pm$0.019) | 0.8795($\pm$0.016) | 0.8872($\pm$0.015) | 0.8833($\pm$0.012) | 0.8832($\pm$0.011) |
| | 5. | Bi-T-LSTM | 0.8401($\pm$0.023) | 0.8811($\pm$0.015) | 0.8945($\pm$0.017) | 0.8871($\pm$0.015) | 0.8946($\pm$0.014) |
| | 6. | Bi-T-VRNN | 0.8358($\pm$0.012) | 0.8812($\pm$0.009) | 0.8845($\pm$0.015) | 0.8817($\pm$0.019) | 0.8856($\pm$0.015) |
| | 7. | Bi-T-Transformer | 0.8564($\pm$0.015) | 0.8774($\pm$0.014) | 0.8884($\pm$0.016) | 0.8812($\pm$0.013) | 0.8954($\pm$0.01) |
| Mayo | 1. | LSTM | 0.8354($\pm$0.01) | 0.8808($\pm$0.013) | 0.8743($\pm$0.014) | 0.8768($\pm$0.013) | 0.8612($\pm$0.011) |
| | 2. | VRNN | 0.835($\pm$0.015) | 0.8727($\pm$0.018) | 0.8712($\pm$0.014) | 0.8719($\pm$0.015) | 0.8582($\pm$0.011) |
| | 3. | Transformer | 0.8455($\pm$0.005) | 0.8781($\pm$0.008) | 0.8755($\pm$0.006) | 0.8757($\pm$0.009) | 0.8609($\pm$0.011) |
| | 4. | RAPT | 0.8452($\pm$0.014) | 0.8841($\pm$0.012) | 0.8815($\pm$0.016) | 0.8828($\pm$0.011) | 0.8782($\pm$0.011) |
| | 5. | Bi-T-LSTM | 0.8467($\pm$0.011) | 0.8887($\pm$0.013) | 0.8842($\pm$0.015) | 0.8851($\pm$0.017) | 0.8819($\pm$0.017) |
| | 6. | Bi-T-VRNN | 0.8423($\pm$0.015) | 0.8872($\pm$0.013) | 0.8795($\pm$0.012) | 0.8823($\pm$0.014) | 0.8824($\pm$0.017) |
| | 7. | Bi-T-Transformer | 0.8481($\pm$0.008) | 0.8854($\pm$0.009) | 0.8857($\pm$0.008) | 0.8855($\pm$0.012) | 0.8843($\pm$0.011) |
| MIMIC | 1. | LSTM | 0.8167($\pm$0.007) | 0.8305($\pm$0.013) | 0.865($\pm$0.011) | 0.8474($\pm$0.014) | 0.8732($\pm$0.011) |
| | 2. | VRNN | 0.8265($\pm$0.013) | 0.8324($\pm$0.013) | 0.858($\pm$0.014) | 0.845($\pm$0.018) | 0.873($\pm$0.016) |
| | 3. | Transformer | 0.8483($\pm$0.009) | 0.847($\pm$0.012) | 0.8384($\pm$0.012) | 0.8427($\pm$0.006) | 0.8775($\pm$0.01) |
| | 4. | RAPT | 0.8512($\pm$0.017) | 0.8592($\pm$0.016) | 0.8632($\pm$0.019) | 0.8612($\pm$0.015) | 0.8802($\pm$0.021) |
| | 5. | Bi-T-LSTM | 0.8299($\pm$0.005) | 0.8512($\pm$0.014) | 0.8673($\pm$0.011) | 0.8592($\pm$0.026) | 0.8842($\pm$0.015) |
| | 6. | Bi-T-VRNN | 0.8267($\pm$0.008) | 0.8634($\pm$0.014) | 0.8572($\pm$0.016) | 0.8602($\pm$0.014) | 0.8821($\pm$0.021) |
| | 7. | Bi-T-Transformer | 0.8573($\pm$0.014) | 0.8675($\pm$0.015) | 0.8592($\pm$0.015) | 0.8633($\pm$0.014) | 0.8834($\pm$0.014) |

fectiveness of combining bidirectionality and time-awareness. These results further support the conclusion that bidirectional and time-aware mechanisms greatly enhance model performance across diverse architectures, outperforming base models like VRNN and Transformer that do not incorporate these enhancements.

Figure 4: AUC for each model based on the early prediction window (MIMIC)

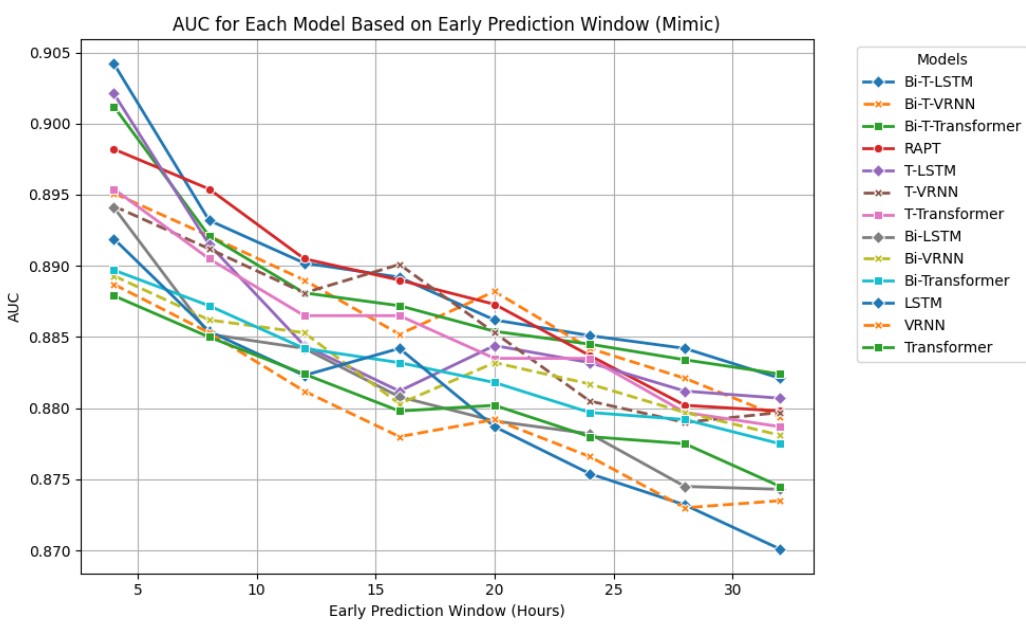

Figure 4, Figure 5, and Figure 6 illustrate the AUC results for each model across various early prediction windows, using the MIMIC, CCHS, and MAYO datasets, respectively. These figures compare the performance of LSTM, VRNN, Transformer, and RAPT models, with the three mechanisms—Bi-Time, Time-Aware, Bi-Directional, and the original versions of each model—across early prediction windows from 4 to 32 hours.The results show consistent trends

Figure 5: AUC for each model based on the early prediction window (CCHS)

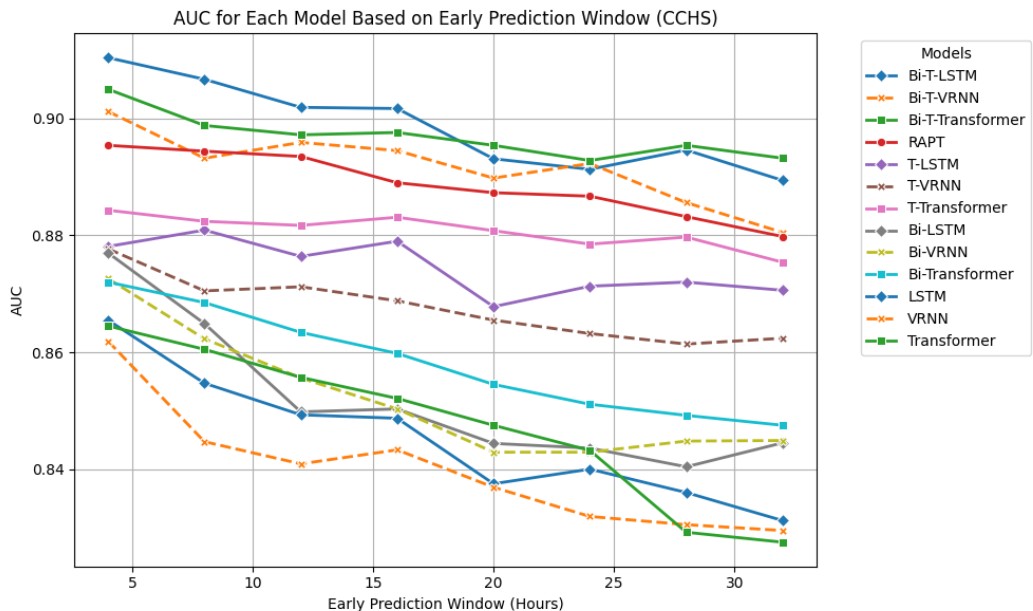

Figure 6: AUC for each model based on the early prediction window (MAYO)

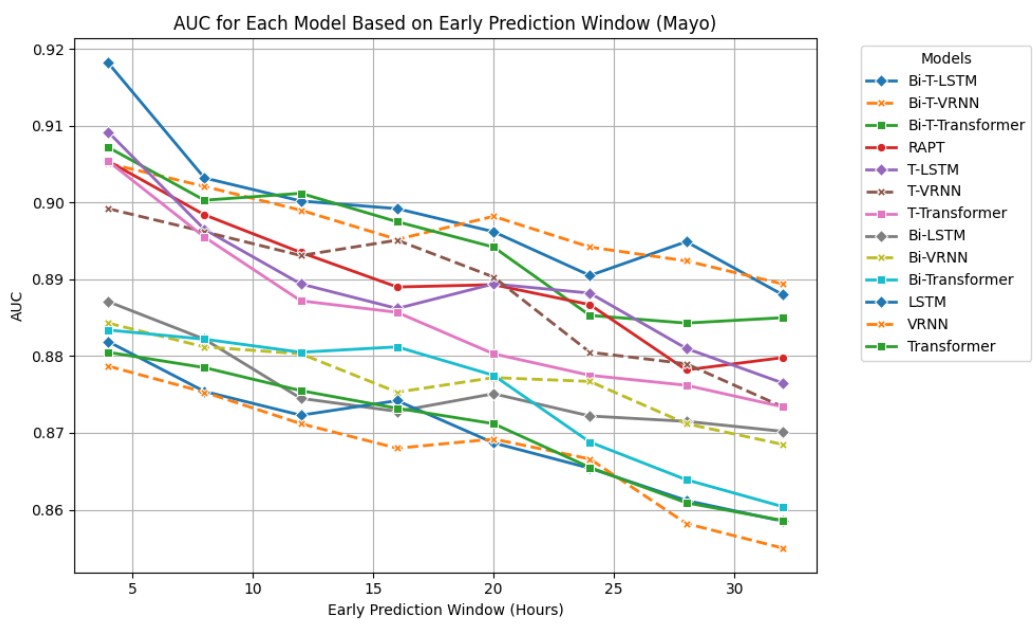

across all three datasets. The Bi-Time mechanism (a combination of bidirectional processing and time-aware encoding) consistently outperforms the other mechanisms for all models and datasets, particularly as the early prediction window increases. The Bi-T-LSTM model, in particular, achieves the highest AUC values in nearly all prediction windows, further reinforcing the strength of bidirectionality and time-awareness in capturing complex temporal dependencies within EHRs.

Figure 7: Critical difference diagram for AUC on the MIMIC Dataset

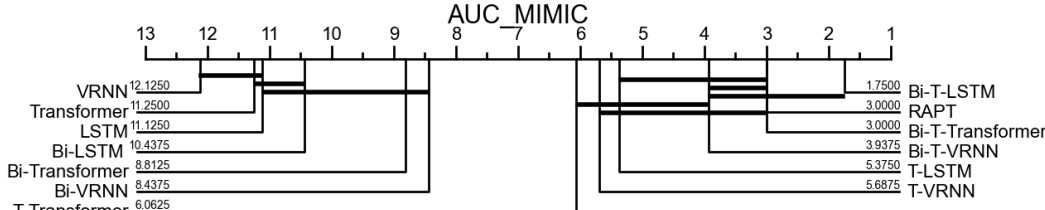

Figure 8: Critical difference diagram for AUC on the CCHS Dataset

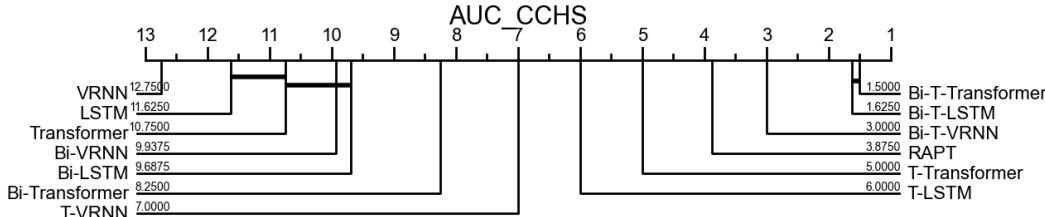

While Bi-T-LSTM leads across most scenarios, VRNN and Transformer models also benefit significantly from the addition of bidirectionality and time-awareness. This shows that the improvements extend beyond the LSTM architecture, demonstrating the general applicability of these mechanisms for early septic shock prediction in complex, time-sensitive datasets.

In Figure 7 and Figure 8, the critical difference diagrams clearly demonstrate the superior performance of bi-directional time-aware models across the MIMIC and CCHS datasets, respectively. As observed, the Bi-T-LSTM consistently ranks as the top-performing model in both datasets, reinforcing its strong predictive capabilities when augmented with bidirectionality and time-awareness.

In the MIMIC dataset (Figure 7), RAPT and Bi-T-Transformer closely follow Bi-T-LSTM, showcasing the broad applicability of these mechanisms across different architectures. Similarly, in the CCHS dataset (Figure 8), Bi-T-LSTM and Bi-T-Transformer outperform other models, further confirming that bidirectionality combined with time-awareness consistently leads to enhanced model performance, regardless of the underlying architecture.

These additional results further strengthen our conclusion that the combination of time-aware encoding and bidirectional processing yields the most powerful predictive models in our study. And further advocates utilizing the fusion of bidirectionality with time-aware mechanisms in the complex, time-sensitive datasets like EHRs to siginifically improve model's performance.

**Code Availabilities and Computational Resources.** Code implementations for all models above can be found in the supplementary materials attached. All experimental workloads are distributed across several Nvidia RTX 2060 6GB GPU clusters