# OpenReview forum: "The Power of Data: How LSTMs Outshine Disease Progression Modeling with Two Simple Mechanisms"
_ICLR.cc/2025/Conference — ICLR 2025 Conference Withdrawn Submission_

### Official Review · Reviewer_xzxP · 2024-10-28

**Soundness:** 2
**Presentation:** 2
**Contribution:** 1
**Rating:** 3
**Confidence:** 4

**Summary:**

In this paper, the authors focus on the area of disease progression modeling by leveraging real-world electronic health records (EHR). They assess the effectiveness of three distinct mechanisms: bidirectionality, time-awareness, and a hybrid approach combining both. These mechanisms are incorporated into three deep learning architectures—LSTM, VRNN, and Transformer models—to evaluate their performance. The findings from experiments conducted on three datasets, namely CCHS, Mayo, and MIMIC, indicate that the hybrid mechanism consistently outperforms the others, delivering the most accurate results across all datasets.

**Strengths:**

* The investigation into disease progression modeling holds substantial medical significance, offering potential benefits to both clinicians and patients by enhancing the accuracy and timeliness of diagnoses.

* The incorporation of bidirectionality, time-awareness, and their combined mechanism into the investigated deep learning models is technically reasonable.

* The experimental evaluation includes a comparative analysis of the proposed mechanisms against the baseline model RAPT, and an exploration of the impact of varying early prediction window lengths on three real-world EHR datasets: CCHS, MIMIC, and Mayo.

**Weaknesses:**

* The research utilizes two existing mechanisms—bidirectionality and time-awareness—and combines them in a relatively straightforward manner, without elaborating on the challenges or innovative aspects of their integration. This, as a result, diminishes the perceived novelty of the proposal.

* The choice to focus on early sepsis shock prediction as the primary application throughout the datasets is questionable given the paper's stated aim of modeling disease progression. This application does not seem representative of the broader domain of disease progression. Including other applications would not only enhance the applicability of the findings but also demonstrate the versatility of the proposed mechanisms in handling different disease trajectories.

* In Section 2.1, the paper integrates time interval information into the models in two ways: (i) as time embeddings in the augmented input and (ii) as a controlling factor in LSTM gates. However, the justification for implementing time intervals in these dual capacities is inadequately detailed. In addition, an ablation study to evaluate the impact of each integration way would clarify the necessity and effectiveness of these design choices.

* The selection of RAPT as the sole baseline for comparison is a limitation. Including a broader range of baseline models, particularly those representing the state-of-the-art in time-series EHR analytics, would strengthen the evaluation. Further, presenting the performance of RAPT across varying early prediction windows would offer a clearer benchmark, thereby highlighting the advantages and robustness of the proposed mechanisms.

* Given the focus on EHR analytics, an interpretability study of the three mechanisms or a clinical validation providing insights into their practical implications would be valuable. Such an analysis would elucidate how the proposed mechanisms might support clinical decision-making in real-world settings, thereby adding substantial value to the research.

**Questions:**

Please check the weaknesses above.

---

### Official Review · Reviewer_xGbs · 2024-10-31

**Soundness:** 2
**Presentation:** 3
**Contribution:** 2
**Rating:** 1
**Confidence:** 5

**Summary:**

This work focuses on improving disease progression modeling (DPM) for electronic health records (EHRs) using Long Short-Term Memory (LSTM), Variational Recurrent Neural Networks (VRNN), and Transformer models. By introducing two mechanisms—bidirectionality and time-awareness—the authors claim to address challenges in EHR data, such as sequence directionality and irregular time intervals. These enhancements help predict critical conditions like septic shock more accurately. Experimental results across three healthcare datasets show that models incorporating both mechanisms outperform standard approaches.

**Strengths:**

- The primary strength of this paper lies in its clear presentation and structured approach to describing the model and experimental process.
- The authors provide detailed, accessible explanations of the bidirectional and time-aware mechanisms, as well as their integration into LSTM, VRNN, and Transformer models.
- The clarity in both the model formulation and experimental design makes the paper more accessible to readers.

**Weaknesses:**

- The novelty of the proposed approaches appears limited. Ma et al. [1] were the first to introduce bidirectional RNNs for disease progression modeling using EHRs, while Baytas et al. [2] initially proposed methods for handling irregular time intervals. Building on these concepts, Hitanet [3] introduced a time-aware Transformer to address disease progression modeling challenges. The Transformer architecture’s inherent bidirectionality, achieved through QKV calculations that allow attention across all tokens, enhances this approach. Additionally, Hitanet accounts for irregular time intervals, as do other methods, such as T-ContextGGAN [4], which also models irregular time intervals using Transformer-based structures. The authors should clarify the differences between the proposed approaches and existing works.
- Several key references are missing. Please refer to [5], a recent survey summarizing existing work on this topic. For example, in Section 3.2 of [5], the authors listed several models that use irregular time intervals.
- The authors only use a few baselines in the experiments. However, as mentioned before, most of the models listed in the survey [5] should be used as baselines, such as Hitanet [3], Diople [1], and others listed in Table 1 of [5].

[1] Ma et al.,  Dipole: Diagnosis prediction in healthcare via attention-based bidirectional recurrent neural networks. SIGKDD, 2017.

[2] Baytas et al., Patient subtyping via time-aware LSTM networks. SIGKDD, 2017.

[3] Luo et al. Hitanet: Hierarchical time-aware attention networks for risk prediction on electronic health records. SIGKDD, 2020.

[4] Xu et al., Time-aware context-gated graph attention network for clinical risk prediction. IEEE TKDE, 2022.

[5] Wang et al., Recent Advances in Predictive Modeling with Electronic Health Records. IJCAI, 2024.

**Questions:**

See weaknesses.

---

### Official Review · Reviewer_sn24 · 2024-11-02

**Soundness:** 2
**Presentation:** 2
**Contribution:** 4
**Rating:** 5
**Confidence:** 5

**Summary:**

The paper addresses Disease Progression Modeling (DPM) in Electronic Health Records (EHRs). It particularly focuses on a deadly phenotype, namely septic shock, which is a critical condition causing high mortality in ICU patients. The study proposes solutions for two key challenges in longitudinal EHR modeling: irregular sampling and the presence of subtle symptoms. To address these, it proposes time-varying embedding and bidirectional context modeling. As a result, these mechanisms could handle the irregular sampling of events and model both historical and future information to improve prediction performance. Because these techniques are generalizable to unidirectional deep learning models, like RNN and Transformer, the authors validate the different combinations of unidirectional models, time-varying mechanism, and bidirectional modeling. The results demonstrate the improvements of these techniques and the potential for future research. Given my past research in sepsis prediction from ICU data, I believe this research addresses a key challenge and  could have significant impact. The inclusion of bidirectional contexts and time-varying information is reasonable for improving model accuracy.

Although the paper presents a well-motivated solution with impactful potential applications in AI for healthcare, it has two crucial limitations that prevent its acceptance. The first main limitation is the poor presentation of technical contributions. It does not clearly distinguish the existing knowledge about RNN/Transformer, and it does not display techinical noveltiy. The second main limitation is that its technical development lags behind research advancements in Linear Transformer, exponential decay, and relative position embedding. These limitations impact the paper's technical contribution.

Overall, this research has strong practical applications in real-world healthcare but could benefit from **better presentation** and **the inclusion of advanced techniques in Related Works**. If the authors address these weaknesses, I would consider adjusting my evaluation.

**Strengths:**

1. This study demonstrates a strong clinical understanding of septic shock, effectively integrating clinical insights into model design. It also explains clinical uses, such as the early detection of sepsis.

2. For EHR data and sepsis, it identifies key challenges of irregularly-sampled time series and the subtle, early symptoms.

3. This study discusses potential solutions based on the motivation as well as the challenges. The proposed bidirectional and time-aware approaches are reasonable ways to address key limitations.

4. The authors clearly explain why the proposed bidirectional and time-aware mechanisms improve model performance over original unidirectional models.

5. One insightful component of this paper is the use of VRNN with time-varying mechanisms incorporated into the latent distribution. This approach emphasizes the sensitivity of VRNN to variable time intervals, which may potentially better reflect temporal dynamics and capture subtle variations in health data.

6. A reliable aspect of this study is the identification of gold standard labels for septic shock, a factor often neglected in favor of simple ICD codes.

7. The experiments leverage a large, multi-institutional ICU dataset. Given the limited access to healthcare data in research, this is an important advantage compared to other studies.

**Weaknesses:**

1. The paper incorrectly uses the term "forecasting",  whereas the experiment focuses on septic shock prediction (i.e., classification).


2. The descriptions of LSTM and bi-LSTM in Section 2 should be merged into a background section. Because they are not technical contributions in this paper. Doing so, it could clearly distinguish the exsting works from the proposed contributions.


3. Section 2 is too complex and nested, where its content could be more concise. For example, Section 2.1 dicusses time-varying embedding in LSTM. However, there is no need for 2.1.1, as the LSTM should be discussed in background as existing research. Moreover, the content in Section 2.1.1 could be simplified significantly. You can first describe the exponential decay function $ \gamma(\Delta \tilde{t}) = \exp(-\alpha \Delta \tilde{t})$ and the exponential decayed hidden states $h^{t-1} \cdot \gamma(\Delta \tilde{t})$. Therefore, this paper does not need to include equations of each LSTM gate (Forget Gate, Input Gate, Cell State Update, Output Gate) using the decay function.


4. Some valuable technical details are currently located in the appendix. For example, the discussion of how to incorporate time-varying mechanism in RNN in appendix A should be moved to the main text. I believe A.5 is particularly important, it has not been discussed before, but is not currently presented in the main text.


5. Some valuable technical discussions are missing. For example, there is no discussion of the learnable decay rate $\alpha$. Explaining its technical meaning and clinical insights (i.e., the connection between decay rate and chronic disease) could add values to this study. Some experiments could be done to support it.

6.  This study may not familiar with the recent technical advancements in relative position embedding (RPE). For example, the time-varying embedding $e^t = E_t(\Delta \tilde{t})$ is a variant of RPE. In Rotary PE (RoPE) [1],  $Q_n^\top K_m = W_Q^\top X_n^\top  e^{-i \theta (n - m)} X_m W_K$, we can treat $\Delta \tilde{t} = n - m$ to make RoPE time-varying.


    Some existing works have discussed the connection between the proposed time-varying embedding and RPE [2]. **The authors should discuss the connection to RPE (RoPE, xPos) in the Related Work section** [1,3].

    [1] RoFormer: Enhanced transformer with Rotary Position Embedding. Neurocomputing, 2024.

    [2] TimelyGPT: Extrapolatable Transformer Pre-training for Long-term Time-Series Forecasting in Healthcare, ACM-BCB 2024.

    [3] A Length-Extrapolatable Transformer. ACL 2020.

7. This study misses the important connection of the proposed exponential decay mechanism and linear Transformer [2]. The linear Transformer rewrites Attention as Linear Attention, which is a Linear RNN model with a state variable. Among Linear Transformers, RWKV and RetNet apply exponential decay to the state variables [3,4]. For example, given the exponential decayed state variables in Retnet ($S_n = \gamma S_{n-1} + K_n^\top V_n$) and yours $h^{t-1} \cdot\exp(-\alpha \Delta \tilde{t})$, we can see a clear connection.

    **Therefore, a discussion of Linear Transformers in Related Works should be included**. Given the connection to RPE as well as Linear Transformer, some works with RPE and Linear Attention are actually your proposed BI-T-Transformer model [6], which could be included into Related Work.

    [4] Transformers are RNNs: Fast Autoregressive Transformers with Linear Attention. ICML 2020.

    [5] RWKV: Reinventing RNNs for the Transformer Era. Arxiv.

    [6] Retentive Network: A Successor to Transformer for Large Language Models. Arxiv.

    [7] Bidirectional Generative Pre-training for Improving Healthcare Time-series Representation Learning. MLHC 2020.

    8.The paper could highlight that these techniques are generalizable to all unidirectional models without considering irregular intervals.

9. The paper would benefit from more qualitative analysis. For example, this study mentions septic shock has subtle symptoms in the early stage and the bidirectionality could improve prediction performance. Therefore, it could include a trajectory analysis in a subject that shows the subtle symptoms as well as its impact on labels. Other qualitative analysis and clinical analysis are encouraged.


10. Regarding Transformer, the experiment designs are not adequate. The Transformer includes unidirectional attention (causal attention from Transformer's decoder or GPT) and bidirectional attention (standard self-attention from Transformer's encoder or BERT). Therefore, **decoder-only GPT should be your baseline, BERT should be your bidirectional Transformer**.


    However, this paper introduces a forward and backward self-attention, and then concatenates them together as final representation. This approach is very similar to ELMO paper [6]. However, the BERT paper discusses this difference, showing that the standard self-attention mechanism inherently models both forward and backward context. Consequently, I think the authors should re-conduct the experiments with GPT and BERT.

    [8] Deep contextualized word representations. NAACL 2018.

    [9] BERT: Pre-training of Deep Bidirectional Transformers for Language Understanding. NAACL 2019.

**Questions:**

I also have a few questions, and I'm unsure whether they represent limitations of the paper or simply my own confusion. Could you please help clarify?

1. What is the detailed detailed setup of various models, like the embedding size of time-varying mechanism the layer number for different models.

2. In Section 3.1, why we need to do mean-fill for the missing values? This model all values available for each timestep, while it can handle missing timesteps?

3. How does the incorporation of time-varying embedding impact on the variational distributions in VRNN?

---

### Official Review · Reviewer_Bb9V · 2024-11-03

**Soundness:** 2
**Presentation:** 3
**Contribution:** 1
**Rating:** 3
**Confidence:** 5

**Summary:**

The paper introduces bidirectional and time-aware mechanisms to improve deep learning models for Disease Progression Modeling using EHR data. The method introduces time embedding modeling to enhance models such as LSTM, VRNN, and Transformers for early prediction of septic shock. Experimental results show significant performance gains across various models and datasets, with Bi-T-LSTM achieving notable success.

**Strengths:**

1. Comprehensive Evaluation: Assesses the mechanisms across multiple models (LSTM, VRNN, Transformer) and datasets, demonstrating robust applicability.

2. Real-World Relevance: Addresses practical challenges in EHR data, such as irregular time intervals, making findings highly applicable to real-world healthcare.

**Weaknesses:**

1. Lack of Novelty and Incomplete Related Work Analysis:

The paper's primary mechanisms—bidirectional encoding and time delta embedding—have been previously explored in the context of EHR data modeling. For instance, T-LSTM [1] effectively uses a time decay function to model time gaps, while HiTANet [2] integrates adaptive time embeddings into the transformer framework for EHR sequence encoding. In fact, the authors already mention the T-LSTM in the related works but fail to realize the proposed idea is highly similar to the paper and fail to provide any comparison or illustration. To strengthen the claim of novelty, I suggest the authors conduct a more thorough review of related works and explicitly outline how their proposed approach differs from or improves upon these prior methods. Including later and related advancements would clarify the unique contributions of this paper and address overlaps with existing research.

2. Insufficient Experiment Details:

While the comparison across different neural architectures is commendable, the paper lacks specific details about model configurations, which are crucial given the typically limited size of EHR datasets. Model parameter size and the number of layers can significantly impact performance, particularly for complex models such as transformers. To enhance the reproducibility and reliability of results, I recommend the authors include detailed information on the parameter sizes, layer configurations, and training strategies of each model tested. This would provide a clearer understanding of how resource constraints might have influenced the observed outcomes and support fair comparisons across models.


Ref
1.  Baytas I M, Xiao C, Zhang X, et al. Patient subtyping via time-aware LSTM networks[C]//Proceedings of the 23rd ACM SIGKDD international conference on knowledge discovery and data mining. 2017: 65-74.

2. Luo J, Ye M, Xiao C, et al. Hitanet: Hierarchical time-aware attention networks for risk prediction on electronic health records[C]//Proceedings of the 26th ACM SIGKDD international conference on knowledge discovery & data mining. 2020: 647-656.

**Questions:**

1. Compared to T-LSTM and Hitanet, what parts of your proposed idea are different?
2. Have the author conducted an experiment to compare the proposed method with previous methods that also focus on the time modeling of EHR data?

---

### Note · Authors · 2024-12-10

I have read and agree with the venue's withdrawal policy on behalf of myself and my co-authors.